# Parathyroid Disease in Pregnancy and Lactation: A Narrative Review of the Literature

**DOI:** 10.3390/biomedicines9050475

**Published:** 2021-04-26

**Authors:** Elena Tsourdi, Athanasios D. Anastasilakis

**Affiliations:** 1Center for Healthy Aging, Department of Medicine III, Technische Universität Dresden Medical Center, 01307 Dresden, Germany; 2Department of Endocrinology, 424 General Military Hospital, 56429 Thessaloniki, Greece; a.anastasilakis@gmail.com

**Keywords:** hyperparathyroidism, hypoparathyroidism, calcium, pregnancy, lactation

## Abstract

Pregnancy and lactation are characterized by sophisticated adaptations of calcium homeostasis, aiming to meet fetal, neonatal, and maternal calcium requirements. Pregnancy is primarily characterized by an enhancement of intestinal calcium absorption, whereas during lactation additional calcium is obtained through resorption from the maternal skeleton, a process which leads to bone loss but is reversible following weaning. These maternal adaptations during pregnancy and lactation may influence or confound the presentation, diagnosis, and management of parathyroid disorders such as primary hyperparathyroidism or hypoparathyroidism. Parathyroid diseases are uncommon in these settings but can be severe when they occur and may affect both maternal and fetal health. This review aims to delineate the changes in calcium physiology that occur with pregnancy and lactation, describe the disorders of calcium and parathyroid physiology that can occur, and outline treatment strategies for these diseases in the above settings.

## 1. Introduction and Methodology

Pregnancy and lactation are characterized by profound changes in calcium physiology, aiming at mineralizing the skeleton of the developing fetus and maintaining calcium homeostasis of the neonate [1,2]. During pregnancy, the fetal demand for this mineral is mainly met through a significant increase in maternal intestinal calcium absorption, which more than doubles. In contrast, during lactation, calcium is derived primarily from the maternal skeleton through the processes of osteoclast-mediated bone resorption and osteocytic osteolysis [2]. These maternal changes during pregnancy and lactation can influence the manifestations, diagnosis, and management of calcium disorders such as primary hyperparathyroidism and hypoparathyroidism [3]. Although they rarely complicate pregnancy, metabolic bone disorders could have dire consequences for maternal and fetal health. Moreover, it remains currently unknown whether more subtle abnormalities in calcium, vitamin D, and parathyroid hormone levels have clinically important long-term effects on mothers and neonates. In the present review, we aim to briefly review physiological changes in calcium homeostasis during pregnancy and lactation and focus on disorders of calcium and parathyroid physiology, as well as presenting existing treatment approaches. 

We searched electronic databases (PubMed/MEDLINE) and ClinicalTrials.gov using MeSH terms “Pregnancy”, “Lactation”, “Primary Hyperparathyroidism”, and “Hypoparathyroidism” up to 28 March 2021. As part of the search for this narrative review, we identified 860 abstracts on PubMed and no clinical trials on ClinicalTrials.gov. After eliminating publications that were not pertinent to the subject of parathyroid and calcium disorders in pregnancy and lactation and duplicates, and extending the literature search by manual searching of the references of selected articles, we retained 100 abstracts on PubMed. In view of the paucity of data, we chose to include case reports and case series with regard to manifestations and diagnostic or therapeutic procedures of a respective disease, but have indicated this limitation within the manuscript.

## 2. Calcium Physiology during Pregnancy and Lactation

During gestation, the average fetus requires approximately 30 g of calcium in order to mineralize its skeleton [1,2]. The majority of calcium transportation from the mother to the fetus takes places during the third trimester across a placental–fetal calcium gradient of 1.0:1.4 [4]. This active calcium transportation is not regulated by maternal parathyroid hormone (PTH), but depends on placental and fetal secretion of PTH-related protein (PTHrP). The enhanced calcium demand in the pregnant state is not primarily met through augmented skeletal resorption of the maternal skeleton, but through increased intestinal calcium absorption. In detail, the amount of calcium absorbed by the mother during pregnancy effectively doubles to reach 400 mg per day [1,2]. This increase in maternal calcium absorption is mediated by an upregulation of 1,25-dihydroxy-vitamin D (1,25(OH)_2_D; calcitriol), the production of which rises significantly through an increase of the activity of 1a-hydroxylase, mainly occurring at the placenta. This procedure is also regulated by PTHrP in the pregnant state (Figure 1). Thus, PTHrP plays a pivotal role in calcium homeostasis in pregnancy and lactation, which is quite distinct from its pathophysiological mechanism of action in the setting of tumor-induced humoral hypercalcemia [2]. The significance of PTHrP in development is emphasized by its broad expression in many fetal tissues [5], but also in its abundant distribution in the placenta, fetal membranes, and amniotic fluid [6,7]. Indeed, the absence of PTHrP expression in mice results in lethality [8], whereas the expression of a truncated PTHrP form results in early postnatal demise [9].

As a result of its close homology with PTH, PTHrP binds to the same PTH-1 receptor (PTH1R) and thus simulates some of the actions of PTH, including increases in bone resorption and distal tubular calcium reabsorption; however, due to structural differences, PTHrP stimulates calcitriol production and subsequent intestinal calcium absorption to a lesser degree than PTH [10].

Normally, PTHrP actions are mostly paracrine or autocrine; however, it exerts endocrine effects during pregnancy and lactation. In these settings, PTHrP is produced by the placenta and the mammary gland during pregnancy or by the mammary gland during lactation [11].

In contrast to PTHrP, PTH falls during pregnancy to a low normal range during the first trimester and may increase back to a mid-normal range by the end of gestation [2].

These physiological adaptations translate into specific biochemical changes in the pregnant individual. In detail, intravascular fluid expansion and ensuing hemodilution cause a decrease in the serum albumin concentration. Because of the high percentage of total calcium being bound to albumin (approximately 50%), total calcium is often spuriously low in pregnant women. By contrast, albumin-corrected calcium and ionized calcium remain unchanged and reliably reflect calcium status. Moreover, increased glomerular filtration results in maternal hypercalciuria. Furthermore, bone turnover markers are increased during pregnancy, and especially during lactation [1,2]. These changes, together with the aforementioned alterations in calciotropic hormones, should be taken into account when differentiating between physiological changes and disorders of calcium metabolism.

While 25 hydroxy-vitamin D (25(OH)D) crosses the placenta via diffusion, 1,25-(OH)_2_D does not, and is primarily produced through fetal metabolism [4]. In fact, 1,25-(OH)_2_D is not necessary for fetal calcium metabolism but is important for calcium homeostasis of the neonate after delivery, as well as for postnatal skeletal mineralization. Of note, after the interruption of placental calcium delivery at birth, neonatal ionized calcium decreases, reaching its lowest concentrations 12 h post-delivery, before reaching the normal postnatal range within 7–14 days, when neonatal parathyroid glands have become fully functional [4].

In contrast to pregnancy, the changes in calcium homeostasis during lactation are primarily due to an increase in net bone resorption and the release of calcium from bones into the systemic circulation, amounting to a daily demand of 300–1000 mg calcium per day [1,2]. Calcium mobilization is mostly achieved through the actions of PTHrP produced by the mother’s mammary tissue, whereas the decrease in estrogen levels, which characterizes lactation, also promotes bone resorption. In addition, PTH during lactation normally remains partially suppressed (because of PTHrP surplus), whereas 1,25-(OH)_2_D levels gradually return to normal. This upregulated bone turnover leads to an average loss of 3% of total bone mass during lactation, although a percentage as high as 10% has been reported [1]. PTHrP has a key role in bone loss in this setting, as was elegantly shown in a rodent model of mammary-specific deletion of PTHrP, which resulted in preservation of bone mass during lactation [12]. In addition, the highest concentrations of PTHrP are found in milk, indicating that milk-based PTHrP may reduce mineral accretion by the newborn skeleton [11]. Of note, lactation-induced bone loss is normally fully reversible after weaning, a feature which differentiates this state from other forms of osteoporosis, although cases of irreversible impairment of bone mass and microstructure have been reported [13,14]. In rare cases, fragility fractures may ensue during pregnancy or lactation, accompanied by a persistently low bone mineral density; this entity is referred to as “pregnancy- and lactation-associated osteoporosis”. A number of recent excellent overviews have been published on this subject [15,16,17], which is outside the scope of this current review.

## 3. Disorders of Calcium and Parathyroid Physiology during Pregnancy and Lactation

### 3.1. Primary Hyperparathyroidism

Primary hyperparathyroidism (pHPT), the most common cause of hypercalcemia, is characterized by hypercalcemia and PTH levels that are either frankly elevated or inappropriately normal. It has a prevalence of 0.15–0.4% in the general population and a female-to-male ratio of 2:1 [18]. The vast majority of patients with pHPT are diagnosed in late middle age, with less than 1% currently diagnosed during pregnancy [19], whereas the background prevalence of pHPT in women of reproductive age is estimated to be 36/100,000 [20]. In a prospective approach, aiming to establish the prevalence of undiagnosed pHPT in recurrent miscarriage, DiMarco and colleagues reported an expected prevalence of 0.34% in pregnant women with recurrent miscarriage [21].

The biochemical changes in calcium metabolism which characterize pregnancy and are highlighted above can obscure the diagnosis of pHPT. In particular, total calcium concentrations can appear spuriously lower and PTH can be physiologically suppressed; nevertheless, the combination of increased ionized or albumin-corrected calcium, and/or hypophosphatemia associated with detectable PTH is indicative of pHPT in most cases [2]. Clinical manifestations can also be unspecific and difficult to differentiate from normal pregnancy, i.e., nausea, vomiting, malaise, and muscle aches. Therefore, a significant proportion of cases may remain undiagnosed [22]. A body of evidence suggests that the clinical presentation and especially the presence of maternal, fetal, and neonatal complications are dependent on the magnitude of hypercalcemia. As such, the earlier literature reports describing incidences of 60% for maternal and over 80% for fetal/neonatal complications [23,24,25], which are not currently considered to represent the norm.

### 3.2. Pregnancy Outcomes in Primary Hyperparathyroidism

With regard to pregnancy outcomes in women with pHPT, most data are derived from retrospective observational studies. Norman et al. described 77 pregnancies in 32 women with pHPT and reported a significant difference in live births between women treated with parathyroid surgery at the second trimester of pregnancy (no pregnancy loss) and women managed conservatively (48% pregnancy loss), although it should be noted that the mean total serum calcium in this cohort amounted to 2.85 mmol/L [26]. Of note, although the study by Norman et al. reported a three-fold risk of miscarriage in women with pHPT, with the rate of fetal loss progressively increasing with maternal serum calcium, sampling bias has to be accounted for, given that participants were already diagnosed with pHPT and selected for parathyroidectomy. Conversely, another study comparing pregnancy outcomes in 74 women with pHPT and a mean corrected calcium of 2.72 mmol/L to 175 normocalcemic women did not identify differences between the two groups regarding miscarriage or other obstetrical complications [22]. In addition, no differences in pregnancy outcomes in women with pHPT who underwent surgery compared with those that did not were reported in the same study [22]. In line with the above, in a Danish registry-based approach, women with pHPT were found to deliver by caesarian section more often than normocalcemic women, but did not have a higher rate of miscarriage or differences in neonatal outcomes [27]. In a retrospective analysis over 15 years (2000–2015) at an obstetric referral hospital, rates of preeclampsia and preterm delivery were higher in patients managed conservatively than in those undergoing surgery. Nevertheless, improved maternofetal outcomes were reported overall in comparison to earlier data [28]. As mentioned above, in the first prospective pilot study in a cohort of women with recurrent miscarriages, DiMarco and colleagues reported one case of pHPT among 289 women (0.34%), which was higher than the 0.05% expected [21]. These data suggest that pregnancy outcomes can depend on population characteristics and the magnitude of hypercalcemia. Thus, larger-scale prospective studies are needed in order to ascertain the true prevalence of pregnancy complications.

### 3.3. Maternal Complications of Primary Hyperparathyroidism

Nephrolithiasis has been classically considered to be the most common finding in symptomatic patients with pHPT during pregnancy [29]. In a review of 70 pregnant women with pHPT, renal colic and objective evidence of nephrolithiasis were found in 17% and 36% of patients, respectively [29]. Pancreatitis constitutes a severe complication, with a prevalence higher than in non-pregnant individuals [30,31,32]. Other clinical findings comprise maternal hypertension and preeclampsia, with some data indicating associations between the presence of parathyroid adenoma and preeclampsia [33]. Severe hypercalcemia or frank parathyroid crisis complicated by cardiac arrhythmias have been reported especially in the post-partum period and can be pathophysiologically explained by an excess of PTHrP during lactation, combined with the loss of placental calcium transfer during pregnancy [34,35]. However, the significant burden of pHPT in terms of maternal complications, as reflected by the earlier literature, should be critically appraised, given that dated methodology may have affected the quality of the reported data.

### 3.4. Fetal Complications of Primary Hyperparathyroidism

Fetal complications of severe hypercalcemia include premature birth, intrauterine growth retardation, and low birth weight [36], as well as neonatal hypocalcemia and tetany because of the suppression of fetal parathyroid tissue, which can persist for weeks after birth [37,38]. Although rare, permanent hypocalcemia has been described as a result of parathyroid aplasia during gestation [39]. The risk of neonatal tetany appears to be proportional to the degree of maternal hypercalcemia. In an earlier case series of 13 women, neonates born to women with a mean peak serum calcium concentration of 3.92 mmol/L developed neonatal tetany, although this was not the case when mean peak maternal calcium concentration was 3.07 mmol/L [40]. However, recent case reports have highlighted this rare, transient fetal complication even in infants born to mothers with asymptomatic pHPT [41,42].

### 3.5. Management of Primary Hyperparathyroidism during Pregnancy

The literature on pHPT in pregnancy primarily consists of case reports and retrospective cohort studies; as a result, no conclusive data exist on whether the risks of conservative management outweigh the risks of surgery in pregnancy. A very recent systematic review evaluated 382 cases of gestational hyperparathyroidism and reported parathyroidectomy during pregnancy in 71.7% and non-surgical management in the remaining 28.3% of the cases [43]. The overall infant complication rate was lower when surgery in the second trimester was performed compared with conservative therapy (9.1% vs. 38.9%). On the basis of this evidence, the authors concluded that parathyroidectomy is associated with fewer risks and better fetal outcomes than more conservative treatments [43]. A number of experts suggest surgery in the second trimester for all pregnant patients with pHPT regardless of the maternal serum calcium concentrations [19,34,44,45,46,47]. Based on a study by Norman and colleagues, who observed that calcium levels higher than 2.85 mmol/L were associated with a particularly high risk of fetal loss [26], some authors recommended surgical treatment when calcium levels are above 2.75 mmol/L, especially for patients with recurrent miscarriages [4,23,48,49,50]. Of note, although parathyroid surgery is known to decrease cardiovascular death rates in the nonpregnant population [51], data on the risk of subsequent preeclampsia attributable to pHPT post-parathyroidectomy are less conclusive; according to one study this risk remains elevated relative to the general population 2 to 5 years after parathyroidectomy [33].

Parathyroidectomy is classically performed during the second trimester because of incomplete organogenesis in the first trimester and the risk of triggering preterm labor in the third trimester, although some reports of uncomplicated surgery in the third trimester have been published [22,47,49,52,53,54,55] (Table 1). If a parathyroid adenoma can be localized preoperatively, minimally invasive parathyroidectomy may be performed using a cervical plexus block, thus avoiding general anesthesia [45,56,57,58,59]. However, preoperative imaging can be challenging in the setting of pregnancy. Although a neck ultrasound can be safely performed in pregnancy and has relatively high specificity, sensitivity is lower and the technique is largely dependent on operator expertise [60,61]. The measurement of PTH in needle wash-out obtained in aspirates of neck lesions has been validated in non-pregnant individuals and has shown high specificity (95–100%) and sensitivity (91–100%) with regard to the identification of parathyroid adenomas [62,63], whereas a recent case report described the successful implementation of this technique in a pregnant woman with marked hypercalcemia [64]. Sestamibi scanning has 80% to 99% sensitivity for the identification of single parathyroid adenomas [65], but 99-Tm-sestamibi is known to cross the placenta [66], so this nuclear medicine technique is avoided during pregnancy. Moreover, CT scanning with multiple contrast phases results in a radiation dose approximately four times higher than sestamibi scanning and should likewise be avoided in pregnancy [55]. Recently, ^18F^fluorocholine and ^11C^mentionine positron emission tomography (PET) have been described as highly sensitive and specific methods for the localization of parathyroid adenomas [67]. To date, a limited number of PET-CT/MRI scans have been performed in pregnant women, almost all for the evaluation of malignancy [68]. Since many imaging modalities are contraindicated in pregnancy, the bilateral surgical approach may be needed to identify all four glands. Intraoperative PTH monitoring to confirm successful excision has been validated in non-pregnant subjects [69], but has also been performed in gestation [4].

**Table 1 biomedicines-09-00475-t001:** Outcomes of surgical and pharmacological management of primary hyperparathyroidism in pregnancy.

Study	Study Design	No of Women/Pregnancies	Intervention	Trimester	Outcome	Comments
Gelister et al., 1989 [70]	Retrospective case series	4	Sx: 2 non-Sx: 2	1st and NR	Sx: uncomplicated non-Sx: 1 uncomplicated; 1 3rd trimester hypertension-CS-healthy infant	
Hsieh et al., 1998 [53]	Retrospective case series	3	Sx: 2 Non-Sx: 1	3rd	Sx: no maternal, fetal, or neonatal complications non-Sx: neonatal hypocalcemia	
Gidiri et al., 2004 [54]	Case series	2	Sx: 2	3rd	healthy infants	
Schnatz & Thaxton 2005 [52]	Review	16	Sx: 16	3rd	Complications 5.9% for fetuses and 0% for mothers	-lower than previously described complications if Sx at 3rd trimester
Truong et al., 2008 [49]	Retrospective case series	3	Sx: 3	2nd and 3rd	No maternal, fetal, or neonatal complications	
Norman et al., 2009 [26]	Retrospective case series	32/77	Sx: 15 non-Sx: 62	2nd	Sx: 15/15 healthy infants non-Sx: 30/62 pregnancy loss	-Pregnancy loss x 3.5 higher -Pregnancy loss at late 1st or early 2nd trimester -Fetal loss associated with maternal Ca levels
DiMarco et al., 2019 [55]	Case series	17	Sx: 15 non-Sx: 2	2nd (*n* = 14) 3rd (*n* = 1)	Sx: 1 CS – 15/15 healthy infants non-Sx: 1 miscarriage – 1 IUGR/CS	
McMullen et al., 2010 [46]	Retrospective case series	7	Sx: 3 non-Sx: 4	2nd	Sx: 3/3 healthy infantnon-Sx: 1/4 pregnancy loss 3/4 preterm delivery	
Abood & Vestergaard, 2014 [27]	Register-based retrospective cohort study	1057/NR	Sx: 576 Non-Sx: 481	2nd and 3rd	no difference in pregnancy outcomes between Sx and non-Sx	-no difference in pregnancy outcomes between women with PHPT and not -more SC in PHPT vs no-PHPT
Hirsch et al., 2015 [22]	Retrospective case series	74/124	Sx: 5	NR	Pregnancy loss 12/124(9.7%) Preterm delivery 2/124(1.6%) Other complications 17/124(13.7%) Sx: 5/5 no maternal or infant complications	-no difference in complications between Sx and non-Sx women -no correlation between maternal Ca and pregnancy outcome
Gokkaya et al., 2016 [71]	Case series	4	Sx: 1 non-Sx: 3		Sx: uncomplicated non-Sx: 1 uncomplicated 1 mother nephrolithiasis/SC 1 preterm SC/infant death	
McCarthy et al., 2019 [50]	Case series	3	Sx: 3	2nd	2 SC – all infants healthy	
Latif et al., 2020 [47]	Case series	2	Sx: 1 non-Sx: 1	3rd	Sx: uncomplicated non-Sx: preterm CS	
Sandler et al., 2021 [43]	Systematic review	382	Sx: 108 non-Sx: 274	Mostly 2nd	Sx: 9/108 infant complications or death—0/108 surgical complications -infant complication rate lower in Sx vs. non-Sx (9.1 vs. 38.9%)	-complications less likely if Sx on the 2nd vs. 3rd trimester

Abbreviations: Ca, calium; CS, cesarean section; PHPT, hyperparathyroidism; IUGR, intrauterine growth restriction; No, number; NR, not reported; Sx, surgery. NB: Table 1 includes case series where n ≥ 2; case reports were not included.

Conservative management comprises intravenously or orally administered rehydration, with or without forced diuresis, as well as pharmacological therapy, although the latter has not commonly been used in pregnancy, mainly due to safety considerations (or lack of data regarding drug safety in pregnancy). However, saline infusion does not have a long-lasting effect, and close maternal and fetal monitoring are warranted to prevent clinical or biochemical deterioration. Calcitonin reduces calcium levels through a direct inhibition of osteoclasts and does not cross the placenta, but its poor effectiveness and a risk of tachyphylaxis impair its use [4,72]. Bisphosphonates, which are often used to treat acute hypercalcemia in non-pregnant individuals, cross the placenta and could be embryotoxic at high doses [73]. In a rodent model, bisphosphonate doses which were eight times higher than those recommended for clinical use resulted in fetal bone malformations [74]. Although the use of bisphosphonates has been reported occasionally without grave side-effects, their regular use is not advised in gestation [75]. Cinacalcet, an activator of the calcium sensing receptor (CaSR), has been used in isolated cases of pregnant women with pHPT with modest success [76]. In a study of 43 non-pregnant individuals with sporadic primary hyperparathyroidism, treatment with cinacalcet was more effective in achieving calcium control in patients not fulfilling criteria for surgical intervention, which could imply that such an effect might be extrapolated in gestational hyperparathyroidism [77]. However, this is just a hypothesis that remains to be proven, and it should be noted that cinacalcet crosses the placenta and the CaSR also regulates fetal–placental calcium transfer [78], so the possibility of adverse effects on the fetus and neonate cannot be excluded [2]. Denosumab also crosses the placenta and resulted in an osteopetrotic-like phenotype in offspring of cynomolgus monkeys and rats [79,80], so its use is contraindicated in pregnancy.

In conclusion, if pHPT is diagnosed in women of reproductive age who are planning to conceive, surgery should be advised pre-conceptionally, given the complexity of the management of hyperparathyroidism during gestation. The risks of pHPT to the mother and fetus appear to increase with biochemical severity; thus, mild elevations of serum calcium (<0.25 mmol/L above the upper limit of normal) may be managed conservatively with the recommendation of definitive surgical treatment postpartum. In patients with moderate to severe hypercalcemia or symptoms, surgery should be performed preferably during the second trimester at reference centers with expertise in the treatment of such patients.

### 3.6. Management of Primary Hyperparathyroidism during Lactation

After delivery, the postpartum loss of placental calcium transfer, along with increased PTHrP production during lactation, may aggravate hypercalcemia in women with pHPT, resulting in some cases in severe symptomatic hypercalcemia [2,15]. PTHrP-induced hypercalcemia during lactation has been reported especially in the setting of gigantomastia and resolved only after mastectomy or bromocriptine administration [81,82,83]. Thus, women with gestational hypercalcemia and gigantomastia should be especially advised against breastfeeding [84]. In addition to recommendations with regard to breastfeeding, surgery should be scheduled during the postpartum period. This is in line with the most recent consensus for the management of pHPT, in which surgery is recommended for all individuals under the age of 50, a recommendation that encompasses all women of reproductive age [85]. In the case that a woman with pHPT does not opt for surgery during lactation, serum calcium should be closely monitored to identify an increase due to the additive effects of mammary PTHrP and circulating PTH [2]. 

### 3.7. Hereditary Syndromes and Other Causes of Hypercalcemia

Hereditary syndromes such as multiple endocrine neoplasias (MEN1, MEN2A, MEN4), familial hypocalciuric hypercalcemia (FHH), hyperparathyroidism jaw tumor (HPT-JT), and familial isolated primary hyperparathyroidism (FIPH) account for approximately 10% of cases of pHPT, with the remaining 90% occurring sporadically, usually as parathyroid adenomas [86]. Genetic testing in pregnant women with pHPT should be considered due to the earlier age of onset of familial parathyroid disorders compared to sporadic disease [86]. Moreover, genetic testing can guide the management of pHPT in pregnancy; as an example, in MEN1 there is a higher risk of multiple parathyroid gland disease, rendering bilateral neck exploration the surgical approach of choice [86].

With regard to FHH, a group of autosomal dominant disorders caused by inactivating mutations of genes affecting the function of CaSR and characterized by chronic and typically asymptomatic mild hypercalcemia in young individuals [87], special attention is needed in pregnancy. Distinguishing between pHPT and FHH in this setting can be particularly challenging due to the changes in calcium physiology described above. In particular, the increased calcium absorption seen in gestation can interfere with the urine calcium/creatinine ratio, which is considered the best test to differentiate FHH from pHPT [88,89,90,91]. Importantly, there is no need for specific treatment for FHH since it is a benign and parathyroidectomy will not restore calcium levels to normal, so past biochemistry, family history and confirmation of the diagnosis with genetic testing can prove very valuable. A small number of cases of FHH in pregnancy have been reported to date [88,89,90,91]. Of note, neonatal biochemical constellations in FHH differ according to genotype. While unaffected neonates can develop hypocalcemia shortly after birth because of parathyroid gland suppression in utero, heterozygous neonates born to an affected mother will be hypercalcemic. Heterozygous offspring of an unaffected mother can develop severe hypercalcemia due to secondary hyperparathyroidism, whereas homozygous neonates are at risk of severe hyperparathyroidism [88,92,93]. Thus, close monitoring of mother and fetus in the pre- and post-partum periods is warranted.

Parathyroid carcinoma is a rare cause of pHPT in the general population, usually accounting for approximately 1% of pHPT in large databases [94]. It has a diverse etiology, comprising neck irradiation, long-standing untreated secondary hyperparathyroidism or genetic causes [95,96,97]. Parathyroid carcinoma is an extremely rare occurrence in pregnancy, with a few cases having been reported to date [98,99,100,101,102,103,104]. This condition is associated with significant neonatal morbidity and mortality, as well as maternal morbidity, and is very difficult to treat. If parathyroid cancer is suspected (based on clinical presentation and cytology), management includes early and complete surgical restriction with negative margins and removal of surrounding lymph nodes to maximize the chance for cure [103].

PTHrP-induced hypercalcemia is a very rare cause of hypercalcemia during pregnancy or lactation and can present with calcium increases of various severities [105]. It is normally characterized by elevated serum calcium and PTHrP and suppressed PTH; the source of PTHrP can be either breast tissue or placenta. PTHrP can be produced by the mammary glands in reaction to prolactin receptor activation, in which case dopamine agonists can be applied to decrease prolactin [81] and to prevent reduction mammoplasty, which has been used in the past [82,83]. In another case, the hypercalcemia, elevated PTHrP, and suppressed PTH were promptly reversed after an urgent cesarean section, thereby indicating a placental origin of PTHrP [106].

Loss of the catabolic effects of 24-hydroxylase due to loss-of-function mutations of the *CYP24A1* gene which encodes the mitochondrial 24-hydroxylase that inactivates 1,25-(OH)_2_D can cause mild hypercalcemia in the non-pregnant state [107]. However, during pregnancy, and because of the unopposed increase in calcitriol, significant hypercalcemia can occur [108], occasionally accompanied by acute pancreatitis [109,110] or late-onset hypertension [111,112].

### 3.8. Hypoparathyroidism

Although total serum calcium physiologically decreases during pregnancy [2], in most cases a healthy and balanced maternal diet, adequate vitamin D intake, and the physiological adaptations of calcium homeostasis during gestation prevent the development of significant hypocalcemia. Thus, causes of clinically relevant hypocalcemia in pregnant women comprise pre-existing hypoparathyroidism, severe dietary calcium restriction, or inadequate vitamin D supplementation, and pseudohypoparathyroidism [113]. The latter refers to a genetic disorder causing resistance to PTH action and presenting with hypocalcemia, hypophosphatemia, and elevated PTH levels [114].

Hypoparathyroidism is characterized by inadequate production of PTH, leading to hypocalcemia and hyperphosphatemia. The most common cause of hypoparathyroidism is thyroid or parathyroid surgery with removal of parathyroid tissue, comprising 75% of cases, with parathyroid aplasia, autoimmune parathyroid tissue destruction, or destruction due to neck irradiation being less frequent causes [115]. The prevalence of hypoparathyroidism in the general population ranges from 0.5% to 6.6% [115]. Of note, since postoperative hypoparathyroidism is most commonly associated with surgery for goiter, thyroid cancer, or Graves’ disease [116], and the median age for these conditions is 45–50 years of age [117,118], postsurgical hypoparathyroidism is rarely encountered in pregnancy as many women conceive in younger years. Indeed, there are no available data on the prevalence of hypoparathyroidism in pregnancy.

### 3.9. Maternal and Fetal Complications of Hypoparathyroidism

Hypoparathyroidism in pregnancy is associated with maternal and fetal complications. In the mother, the signs and symptoms of hypocalcemia depend on the severity and rapidity of development, i.e., the rate of decline of serum calcium, duration of hypocalcemia, and adequacy of substitution. Insufficiently managed maternal hypoparathyroidism can cause preterm labor, miscarriage, and stillbirth [119]. Mineralization of the endochondral fetal skeleton largely occurs in the third trimester of pregnancy [120], and even in conditions of shortages of maternal calcium the placenta will extract calcium to meet fetal mineral accrual at the expense of the maternal skeleton [121]. Consequently, fetal hypocalcemia only occurs in cases of severe maternal hypoparathyroidism/hypocalcemia. As a counterbalancing mechanism, fetal parathyroid glands are overtly stimulated and fetal hyperparathyroidism may ensue, leading to demineralization of the fetal skeleton [122,123]. Severe cases present with bowing of the long bones, osteitis fibrosa cystica, intrauterine fractures, low birth weight, and even fetal death [124,125,126].

### 3.10. Management of Hypoparathyroidism during Pregnancy

The majority of data regarding the management of hypoparathyroidism in pregnancy and lactation used to be derived from case series and case reports [127,128,129,130,131,132,133,134], although a recent position paper issued evidence-based recommendations for management [135]. Although some of the abovementioned reports have demonstrated that some women show reduced symptoms and decreased calcium and calcitriol requirements in gestation (attributed to the effect of PTHrP and the increased placental 1α-hydroxylation) which implies a limited role for PTH in the pregnant woman, some others attest to the opposite. Further complicating matters, the physiological decline in total serum calcium due to hemodilution and hypoalbuminemia in pregnancy has at times been misinterpreted as worsening hypocalcemia, resulting in treatment solely based on laboratory values and not on clinical presentation [2,119,130,136,137,138]. Thus, it appears prudent to measure ionized calcium and closely monitor (i.e., every 3–4 weeks) calcium concentrations, aiming for the low normal reference range in order to avoid adverse effects on the development and function of the fetal parathyroid glands [135]. Calcium, calcitriol, and vitamin D supplements can be safely used in pregnant women. Thiazide diuretics should be discontinued in pregnancy, whereas PTH supplementation therapy has not been adequately studied in this setting [135], with a sole case-report of continuous subcutaneous recombinant parathyroid hormone (PTH) (1–34) infusion in a pregnant woman [139]. Interestingly, the needs for changes in treatment doses of active vitamin D and calcium do not follow a specific pattern during pregnancy. High doses of calcitriol and calcium supplementation should generally be avoided since maternal hypercalcemia can result in a number of complications, as outlined in the section on pHPT. Calcitriol has a shorter half-life in comparison to other active vitamin D formulations; thus, in the case of overtreatment, hypercalcemia can resolve earlier [127]. In the case that alfacalcidol (1α-hydroxycalciferol) is used, it should be noted that calcitriol is approximately twice as potent as alfacalcidol [140]. Although the requirements for exogenous calcitriol vary during pregnancy, a substantial decrease in dose (or even discontinuation) is warranted during lactation, whereas a case report of complete withdrawal of supplemental calcium and calcitriol has also been described [137]. This effect appears to be regulated by the increasing levels of PTHrP in the maternal circulation in later pregnancy and post-partum, which may compensate for PTH deficiency [137].

### 3.11. Management of Hypoparathyroidism during Lactation

A continuation of the same calcitriol regimens can lead to post-partum maternal hypercalcemia [141,142], although, interestingly, one case report demonstrated a transient interval of an increased requirement for calcitriol immediately after delivery and before lactation was underway [131]. Calcium should be monitored every 4–6 weeks during lactation and weaning to ensure normocalcemia, and calcium dose requirements usually decrease in breastfeeding mothers due to the increase in mammary PTHrP [133,135]. Importantly, expecting and lactating women should be instructed as to the signs and symptoms of hypocalcemia and hypercalcemia and encouraged to seek medical advice in the case of their occurrence [131,133].

## 4. Conclusions

Maternal adaptations during pregnancy and lactation may influence or confound the presentation, diagnosis, and management of parathyroid disorders such as primary hyperparathyroidism or hypoparathyroidism. Parathyroid diseases are uncommon in these settings but can be severe when they occur, and call for the consideration of both maternal and fetal health. Most data on diagnosis and treatment derive from isolated case reports and case series, leading to low level of evidence and confidence in recommendations, especially with regard to management (Figure 2). An interdisciplinary approach to care, combing the expertise of endocrinologists, gynecologists, pediatricians, and nursing staff is advised. There is a need for prospective studies on parathyroid diseases during pregnancy and lactation in order to improve the quality of available care, although setting up such studies seems quite challenging, given the rarity of these diseases in pregnancy and lactation.

## Figures and Tables

**Figure 1 biomedicines-09-00475-f001:**
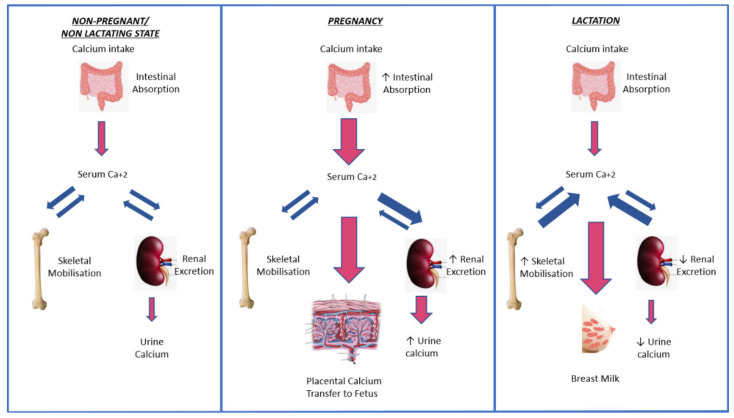
Physiological calcium changes during pregnancy and lactation. Doubles arrows denote reciprocal relationship which single arrows denote increase and decrease.

**Figure 2 biomedicines-09-00475-f002:**
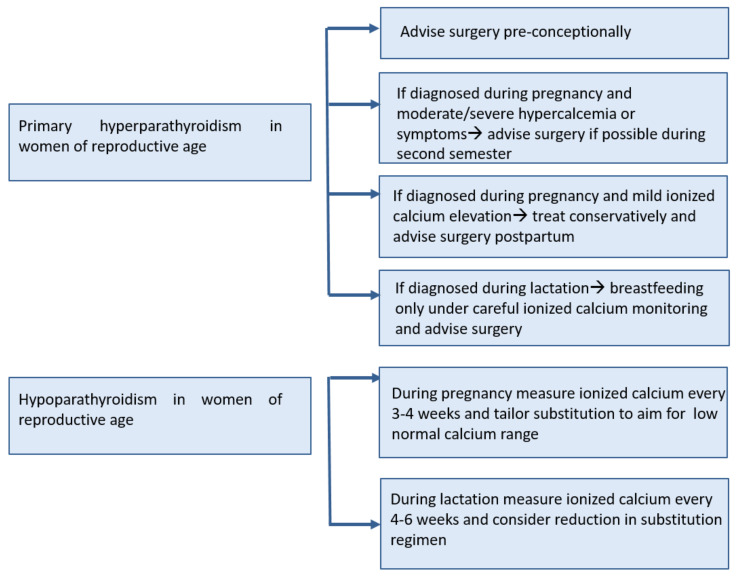
Management of parathyroid disorders during pregnancy and lactation.

## Data Availability

Data were retrieved through a search of electronic databases (PubMed/MEDLINE).

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
