# Peer review of "Parathyroid Disease in Pregnancy and Lactation: A Narrative Review of the Literature"

_biomedicines, 2021, doi:10.3390/biomedicines9050475_

Round 1

Reviewer 1 Report

This review aims to describe the changes in calcium physiology in pregnancy and lactation, focusing on parathyroid diseases in these settings. It is an interesting topic, but available data should be analyzed with more precision and accuracy. Furthermore, the description of parathyroid disease during lactation should be expanded.

A large amount of data come from case series and case reports, as pointed out in the text, but authors should improve their effort to put them together with a stimulating order in the text. Some data and some studies are more important and more significant than others, this cannot be appreciated here. Additionally, some references are very old (22, 25, 33) and it should be highlighted, as it may affect the quality of data.

Paragraph 2 is mainly based only on the first 2 references. Some details about the role of PTHrp should be added, as well as some epidemiological data about pregnancy- and lactation-associated osteoporosis, since it can be a complication of pHPT.  I would suggest to further divide the paragraph “primary hyperparathyroidism” into: a) maternal complications, including nephrolithiasis, pancreatitis, cardiac arrhythmias, hypertension and peptic ulcers b) fetal complications, c) pregnancy outcomes, d) management. The occurrence of maternal and fetal complications should be detailed. Data about miscarriage should be discussed more in depth (lines 121-122), as well as the estimated proportion of undiagnosed case of pHPT (lines 129-130). The risk of neonatal tetany should be reported with more recent available data (Çakır U, Alan S, Erdeve Ö, Atasay B, Şıklar Z, Berberoğlu M, Arslan S. Late neonatal hypocalcemic tetany as a manifestation of unrecognized maternal primary hyperparathyroidism. Turk J Pediatr. 2013 Jul-Aug;55(4):438-40. PMID: 24292040. Pieringer H, Hatzl-Griesenhofer M, Shebl O, Wiesinger-Eidenberger G, Maschek W, Biesenbach G. Hypocalcemic tetany in the newborn as a manifestation of unrecognized maternal primary hyperparathyroidism. Wien Klin Wochenschr. 2007;119(3-4):129-31. doi: 10.1007/s00508-006-0748-1. PMID: 17347863.) instead of the only reference (n°33) of 1964.

Another major limitation of this review is the lack of critical analysis of the data reported, especially regarding the management of hyperparathyroidism, where other references could be added and discussed (Marotta V, Di Somma C, Rubino M, Sciammarella C, Del Prete M, Marciello F, Ramundo V, Circelli L, Buonomano P, Modica R, Vitale M, Colao A, Faggiano A. Potential role of cinacalcet hydrochloride in sporadic primary hyperparathyroidism without surgery indication. Endocrine. 2015 May;49(1):274-8. doi: 10.1007/s12020-014-0381-0. Epub 2014 Aug 15. PMID: 25123977. Latif A, Gastelum AA, Farhan K, Jagadesh S, Mutnuri S. Treatment approach for primary hyperparathyroidism in pregnancy. Proc (Bayl Univ Med Cent). 2020 Oct 28;34(1):191-193. doi: 10.1080/08998280.2020.1834791. PMID: 33456198; PMCID: PMC7785157. McCarthy A, Howarth S, Khoo S, Hale J, Oddy S, Halsall D, Fish B, Mariathasan S, Andrews K, Oyibo SO, Samyraju M, Gajewska-Knapik K, Park SM, Wood D, Moran C, Casey RT. Management of primary hyperparathyroidism in pregnancy: a case series. Endocrinol Diabetes Metab Case Rep. 2019 May 16;2019:19-0039. doi: 10.1530/EDM-19-0039. Epub ahead of print. PMID: 31096181; PMCID: PMC6528402. Truong MT, Lalakea ML, Robbins P, Friduss M. Primary hyperparathyroidism in pregnancy: a case series and review. Laryngoscope. 2008 Nov;118(11):1966-9. doi: 10.1097/MLG.0b013e318180276f. PMID: 18758377). Data about surgical risk should be added and outcomes should be more clearly described. How “scarcely” (line 216) is pharmacological therapy used?

Hereditary syndromes and other rare causes of hypecalcemia could be discussed together.

There is no innovative or provocative proposal: in particular, the reader will be interested to know the real impact of parathyroid diseases during pregnancy and the best proposed management. In this light a summarizing table of the different treatment options could be useful.

The prospective study included in the conclusion (NCT03375359) is not focused on parathyroid disease as mistakenly reported. Indeed it regards first trimester screening for trisomy 21, 18, 13 and 22q11.2 deletion syndrome, the latter having heterogeneous presentation and many possible signs and symptoms that can affect almost any part of the body, including multiple congenital anomalies and later-onset conditions, such as palatal, gastrointestinal and renal abnormalities, autoimmune disease, variable cognitive delays and psychiatric illness.

Revision of the text is required to improve writing and eliminate errors in syntax and punctuation.

Figure 1 can be improved.

Author Response

Reviewer 1:

General Comments: This review aims to describe the changes in calcium physiology in pregnancy and lactation, focusing on parathyroid diseases in these settings. It is an interesting topic, but available data should be analyzed with more precision and accuracy.

We thank this reviewer for her/his careful evaluation.

 Specific Comments:

  1. The description of parathyroid disease during lactation should be expanded.

We have expanded the section of parathyroid disease during lactation.

  1. Some data and some studies are more important and more significant than others, this cannot be appreciated here. Additionally, some references are very old (22, 25, 33) and it should be highlighted, as it may affect the quality of data

We agree with the reviewer and have endeavored to highlight the most significant studies and point out the limitations pertaining older studies in the revised manuscript.

  1. Some details about the role of PTHrp should be added.

We have now added some additional information on the role of PTHrP.

  1. Some epidemiological data about pregnancy- and lactation-associated osteoporosis, since it can be a complication of pHPT.

We have mentioned ‘pregnancy-and lactation-associated osteoporosis’ as a complication of pregnancy/ lactation. However, we believe that it is outside the scope of the current review to extensively elaborate on this entity and have provided references of recent excellent reviews instead.

  1. I would suggest to further divide the paragraph “primary hyperparathyroidism” into: a) maternal complications, including nephrolithiasis, pancreatitis, cardiac arrhythmias, hypertension and peptic ulcers b) fetal complications, c) pregnancy outcomes, d) management.

We have restructured the paragraph as suggested by the reviewer.

  1. Data about miscarriage should be discussed more in depth (lines 121-122), as well as the estimated proportion of undiagnosed case of pHPT (lines 129-130).

We have expanded on data about miscarriage at the respective paragraph of ‘Pregnancy outcomes’. With regard to the estimated proportion of undiagnosed cases of pHPT, to our knowledge, the study of DiMarco et al. (PMID: 29349485), is the most recent prospective study aiming to establish the prevalence of undiagnosed pHPT in the setting of recurrent miscarriage.  

  1. The risk of neonatal tetany should be reported with more recent available data.

We thank the reviewer for highlighting these recent data on the subject of neonatal tetany, which we have now incorporated in the revised manuscript.

  1. Another major limitation of this review is the lack of critical analysis of the data reported, especially regarding the management of hyperparathyroidism, where other references could be added and discussed.

We have added and discussed additional references pertaining the management of hyperparathyroidism. In addition, a very recent systematic review on the management of gestational hyperparathyroidism was included (PMID: 33751589).

  1. Data about surgical risk should be added and outcomes should be more clearly described. How “scarcely” (line 216) is pharmacological therapy used?

We have drafted a table (Table 1) to highlight the outcomes of different modalities of management of primary hyperparathyroidism.

  1. Hereditary syndromes and other rare causes of hypecalcemia could be discussed together.

These subjects are discussed together in the revised version of the manuscript.

  1. There is no innovative or provocative proposal: in particular, the reader will be interested to know the real impact of parathyroid diseases during pregnancy and the best proposed management. In this light a summarizing table of the different treatment options could be useful.

We have drafted a figure (Figure 2) to summarize the different treatment options of parathyroid disorders during pregnancy.

  1. The prospective study included in the conclusion (NCT03375359) is not focused on parathyroid disease as mistakenly reported. Indeed it regards first trimester screening for trisomy 21, 18, 13 and 22q11.2 deletion syndrome, the latter having heterogeneous presentation and many possible signs and symptoms that can affect almost any part of the body, including multiple congenital anomalies and later-onset conditions, such as palatal, gastrointestinal and renal abnormalities, autoimmune disease, variable cognitive delays and psychiatric illness.

We thank the reviewer for drawing this to our attention and have now removed this study.

  1. Revision of the text is required to improve writing and eliminate errors in syntax and punctuation.

We have carefully revised the text as suggested by the reviewer.

  1. Figure 1 can be improved.

We have revised Figure 1.

Reviewer 2 Report

The manuscript is well written and is important for the readers with interest in the subject. My only recommendation is that the authors should specify the deranged calcium metabolism specifically pertaining to pregnancy versus lactation. This maybe incorporated to figure 1 or in a seperate figure. I also recommend the authors provide a table with respect to surgical and pharmacological treatment of primary hyperparathyriodism during pregnancy indicating the number of subjects, trimesters of pregnancy that diagnosis was made and also outcomes.

Author Response

Reviewer 2:

The manuscript is well written and is important for the readers with interest in the subject. My only recommendation is that the authors should specify the deranged calcium metabolism specifically pertaining to pregnancy versus lactation. This may be incorporated to figure 1 or in a separate figure. I also recommend the authors provide a table with respect to surgical and pharmacological treatment of primary hyperparathyroidism during pregnancy indicating the number of subjects, trimesters of pregnancy that diagnosis was made and also outcomes.

We thank this reviewer for her/his positive evaluation. We have revised Figure 1 to highlight the differences in calcium metabolism in pregnancy versus lactation and have drafted Table 1 to summarize outcomes of surgical and pharmacological treatment of primary hyperparathyroidism.

Round 2

Reviewer 1 Report

Substantial changes have improved the overall quality of this paper.

Was table 1 with outcomes of different modalities of management of PHPT included?

In figure 2 second semester should be trimester, please correct.